# Knowledge, Attitudes and Expectations of Physicians with Respect to Palliative Care in Ecuador: A Qualitative Study

**DOI:** 10.3390/ijerph17113906

**Published:** 2020-05-31

**Authors:** Viviana Dávalos-Batallas, Vinita Mahtani-Chugani, Carla López-Núñez, Víctor Duque, Fatima Leon-Larios, Maria-de-las-Mercedes Lomas-Campos, Emilio Sanz

**Affiliations:** 1Department of Health Sciences, Universidad Técnica Particular de Loja (UTPL), Loja 110107, Ecuador; 2Research Unit of the Nuestra Señora de Candelaria University Hospital and Management Office for Primary Care of Tenerife, 38010 Santa Cruz de Tenerife, Spain; vmahchu@gobiernodecanarias.org; 3Department of Psychology, Universidad Loyola Andalucía, 41012 Seville, Spain; clopezn@uloyola.es; 4Úbeda Health Center, Jaén Northeast Health District Andalusian Health Service, 23400 Jaén, Spain; victorduquetf@gmail.com; 5Nursing Department. Faculty of Nursing, Physiotherapy and Podiatry, University of Seville, 41004 Seville, Spain; mlomas@us.es; 6Pharmacology and Physical Medicine Department, University of La Laguna, 38200 San Cristóbal de La Laguna, Spain; esanz@ull.edu.es

**Keywords:** palliative care, Ecuador, medical education, competency framework, qualitative research

## Abstract

*Background:* The worldwide need for palliative care is high, especially in mid- income countries like Ecuador, where the percentage of patients receiving such care is very small due to the scarcity of infrastructure and specialized personnel and to the unequal distribution in the country. The objective of this study is to explore the knowledge, attitudes and expectations related to palliative care of the physicians in Ecuador. *Methods:* A qualitative study based on 28 semi-structured interviews, from March 2014 to November 2016, with physicians working in four cities in Ecuador recruited through the snowball technique. Thematic analysis was developed supported by the ATLAS.ti software. *Results:* Five core themes were identified: (1) training, (2) health policy, (3) professionals’ activities, (4) health services and (5) development of palliative care in Ecuador. *Conclusions:* Strategies are needed which intensify the training of medical professional in palliative care, as well as avail the human resources and materials for providing it.

## 1. Introduction

Palliative care (PC) arises with the intention to provide quality of life to patients facing incurable, life-threatening diseases, and to their relatives by means of providing holistic care which includes psychological and spiritual support, particularly in the moments close to the end-of-life, and during the death and mourning [1]. For this reason, such care is to be performed by a team of professionals that can include physicians from different specializations, nurses, social workers, physical, occupational and speech therapists, nutritionists, pharmacologists, psychologists, voluntary personnel and spiritual counselors (depending on the different religious beliefs), among others. Each one of them contributes their specific competences for a common goal, representing and seeking for a unified care model [2]. The team’s composition shall vary according to the patient needs and to those of their families, to the care model to be implemented and to the available resources [3]. Since approximately two decades ago, in some countries PC programs have been developed, also with the establishment of a networked care which integrates not only different inter-professional teams, but also clinical networks involving the hospital setting and the primary care services [4], thus guaranteeing continuity of care.

According to the World Health Organization (WHO), the worldwide need for PC is high and will continue rising, as a consequence of the ever increasing load that non-transmissible diseases represent and of the aging process of the population [5]. On the other hand, there is evidence in various literature reviews that early referral of the patients to PC determines that they manage their symptoms with more confidence and approach their end-of-life with hope, since it helps them to drive the adaptation process to their diseases and to better understand them [6]. Likewise, the study by Qureshi et al. states that starting PC in the early stages of a disease is associated with a lower use of hospital care, and to an improvement in the quality of the care provided to the patient and of the clinical results of care [7], thus the importance of having units specialized in this type of care.

The Worldwide Hospice Palliative Care Alliance (WHPCA) estimates that at least 40 million people need PC annually and in the world (20 million at the end-of-life stage); however, PC is only integrated in 20 of the 234 countries of the world (8.5%) and 75% of the world population does not have adequate access even to medications to control pain. The biggest development of PC has been recorded in developed countries. Nevertheless, 80% of the individuals who need PC live in low- to mid- income countries [8] like Ecuador that is considered a mid-income country. Despite being the only realistic and human alternative to the suffering of many people, the percentage of patients receiving such care is very low, given the scarcity of infrastructure and of specialized personnel and to the unequal distribution in the country [9]. PC is related to holistic care, where comfort, spiritual care and the end-of-life attention are important premises since their initial models. The PC expansion evidenced by the exponential grows in services worldwide ranged from the attention to patients with cancer to patients suffering non-malignant conditions. In this paradigm challenge, healthcare professionals will need be trained and words like hope, dignity, compassion, attitude and dialog are needed being incorporated into medical studies programs [10].

PC can be provided at the hospital, outpatient and home levels. It is necessary to have hospitalization units specialized in PC [11]. The hospital support groups provide support to the physicians who refer the patient and are in charge of giving recommendations on primary care to the physician [12]. At that level we find the home care teams which allow the patients to end their days at their homes and considerably reduce hospital expenses [13]. Likewise, there are [14] community institutions called hospices for when home care is not possible and hospital admission is not advisable. Consequently, PC shall be provided according to the principles of universal health coverage, as established by the WHO. To such end, training the professionals is fundamental because offering PC should be considered as an ethical duty of the health professionals [5].

The development of palliative care in Ecuador was carried out based on the Spanish model of palliative care. In Spain, The Spanish society of palliative care (SECPAL (Sociedad Española de Cuidados Paliativos), in Spanish) was launched in 1992. The Spanish national palliative care program was developed in 2001. On the other hand, the Spanish national palliative care strategy was developed in 2007, publishing as well, the standards for palliative cares guidelines [15].

The origins of palliative care in Ecuador was associated with a religious congregation of the Camillian Fathers, who established the first hospice that provided palliative care in Quito, Ecuador. After this, The Ecuadorian Federation of palliative care (FECUPAL (Federación Ecuatoriana de Cuidados Paliativos), in Spanish) was founded in 1997. Subsequently, palliative care units in hospitals were opened in Quito and Guayaquil called SOLCA (Hospital de la Sociedad de Lucha contra el Cáncer). The Ministry of Public Health adapted the Spanish guidelines to the context of Ecuador in 2014 [16].

Therefore, the objective of this study is to analyze the situation of PC in Ecuador from the perspective of the physicians in order to elaborate recommendations for the development of PC in the country.

## 2. Materials and Methods

### 2.1. Study Design and Participants

A qualitative study was designed from March 2014 to November 2016 in four Ecuadorian cities: Loja, Quito, Guayaquil, and Cuenca. Quito and Guayaquil are the most important cities in the country, and Loja and Cuenca are smaller cities. This provided us a wide vision of the situation of PC in distinct areas around the country. PC in these cities is similar because physicians follow national politics although access to resources is quite different, having less availability of them in Cuenca and Loja. In addition, these three cities: Guayaquil, Quito and Cuenca show the highest mortality rates in the country.

Semi-structured interviews were conducted with medical professionals living in the different participating cities. The objective was to explore their knowledge, attitudes and expectations as regards PC in order to identify weaknesses and strengths, which later can be translated into health policies of effective coverage for patients susceptible to PC.

### 2.2. Selection of the Participating Sample

The selection of the participants was made by convenience sample and heterogeneity (gender, age, professional experience, place of professional practice and medical specialization) as well as intra-group homogeneity criteria were followed. The subjects were selected in a fair way, without room for prejudices or social, racial, sexual, economic, politic or cultural preferences. The inclusion criteria established were the following: physicians (a) with specialties aligned to PC who practiced Medicine in Ecuador’s public or private sectors; (b) who were able to provide detailed information of their experience related to the theme under investigation, and (c) who wished to participate.

The sample design met the principles of flexibility and reflexivity. In addition, the saturation criterion was followed, since the intended questions were answered, and enough information was obtained [17].

The semi-structured interviews were conducted by the lead author of this article after receiving the pertinent previous training based on knowing phases of the interview and how to handle interviews to achieve this purpose. The script collected in Table 1 was followed, based on the previous literature and agreed upon by the research team. Open questions were used which allowed deepening on the meaning and the experience the participants attributed to the questions proposed.

### 2.3. Procedures

The participants were invited to participate when they met the inclusion criteria by e-mail o by telephone contact. Subsequently, the appointment was made to conduct the interview. The place where the interviews were conducted was agreed upon between the interviewer and the interviewee and it was preferably alien to their workspace not to disturb their routine. The interviews were carried out face-to-face mainly by the lead author, with a mean duration of 60 min and by another member of the research team who took down notes of the session’s progress.

### 2.4. Data Analysis

All the interviews were recorded and later transcribed for analysis by means of the ATLAS.ti Software 7.5.10 (Berlin, Germany). Each participant was assigned a code that allowed guaranteeing his anonymity during the study. After being transcribed, all the interviews were read by the research team in order to guarantee the precision of the statements, as well as the inconsistencies they managed to identify. An inductive approach was followed, with thematic analysis performed to identify the key issues derived from the analysis process. The established steps of global reading, code identification, comparison and contrast of categories were followed to identify similitudes and differences. The codes were grouped in themes to group the texts in their contexts [18].

### 2.5. Ethical Aspects

This study was approved by the Ethics Committee of Research involving Human Beings (*Comité de Éti*ca *de Investigación en Seres Humanos*, CEISH) of the Private Technical University of Loja (*Universidad Técni*ca *Particular de Loja*, UTPL). Prior to signing the informed consent, the participants were informed of the research objectives and goals.

Participation in the study was voluntary, and the individuals could retire from the research at any moment without negative consequences. If the informants decided not to participate any longer, they only had to inform some team member for recording their motivations do so. All information provided by participants was treated with a maximum degree of confidentiality and was used solely for the purposes derived from this research.

### 2.6. Risks for the Participants

No evident risks derived from participating in this study are described.

## 3. Results

### 3.1. Sociodemographic Characteristics of Participants

Initially, 40 physicians were contacted who met the study inclusion criteria. Of these, a total of nine refused to participate due to lack of time or availability. Later on, three interviews were also rejected due to technical problems during the recording. Therefore, a total of 28 semi-structured one-on-one interviews were finally included in the study.

The participants were 13 men and 15 women from the four cities under study, with ages ranging from 34 to 67 years old more than half of the participants were from Quito (15 interviewees), and the rest lived in the cities of Loja (7), Guayaquil (3) and Cuenca (3). The intention was to cover a wide range of professional experience, from 2 to 38 years in more than 10 specializations related to pain management and to PC, among which oncology (10) stands out, followed by anesthesiology (5) and by family medicine (4). Apart from the general training, the one received in relation to PC was taken into account, ranging from attendance to conferences and courses to completing the master’s degree in PC, as well as studies derived from its specialization. Membership of PC associations was also investigated, with our finding that 5 of the interviewees belonged to some organization. The sociodemographic characteristics of the participants can be seen in Table 2.

### 3.2. Analysis Categories in the Interviews

For the analysis of the interview answers, an independent inductive coding was implemented. In this case, the interviews generated a total of 5 categories, as shown in Table 3.

The identified categories supported with the interviewees literal statements are detailed below.

#### 3.2.1. Training

Lack of post-graduate training brings about several consequences, according to the interviewees’ statements. It is observed that the generalists who graduate from the universities to act as rural doctors or as primary care physicians in the health centers and sub-centers are not qualified to identify when a patient is a subject for PC, are unaware of the clinical management of these individuals, and do not know how to communicate with them and with their families.
*“…The sub-centers, which would be the centers where they could help people who don’t have access, the staff is not prepared. So the rural doctor should know palliative care”* (P22).

*“You can’t send any generalist to provide palliative care, because they don’t even know how to approach the patient, how to tell him anything, how to tell the family, that is, everything I’ve learned, trained by the University, of what that means, they don’t have it here. Very few people have it and those who do are pretty empirical, they’re trained, they don’t have the contact, they haven’t lived what they make us do out there. Then, it’s different, then, if you tell me they have so many people by district to care for these patients I don’t know if it is that they are trained”* (P23).

Likewise, the deficit in PC training extends to the research realm, since there are no published papers.
*“Because I’m seeing if there’s any pertinence in palliative care and there’s nothing in Ecuador, because there’s nothing!”* (P7).


A result of the absence of PC in the University curricula and of the lack of knowledge on this specialization is lack of accreditation of the medical professionals’ degrees by the official bodies. This circumstance constitutes an impediment for opening a master’s degree on PC.
*“From the academic practice itself there are problems because from the pre-graduate courses there’s no awareness and, for the post-graduate courses, post-graduate, they require someone with a specialization to come who has super-titles and the only thing they achieved is that we’re in 2016 and up to now there’s no master, or nothing.”* (P16).

Some of the interviewed professionals stated that the concept of PC as a specialization has not been consolidated in the country. The participating physicians highlight the importance of defining and disseminating the concept of PC. The concept not being clear, as they state, there are physicians who call themselves palliative care professional when they are solely devoted to managing one symptom, without having any formal degree or being part of an interdisciplinary team focused on providing an integral treatment to the patient.
*“I believe that it’s an important deficit we have in the country in palliative care, many people think that palliative care is a synonym to pain management and evidently that’s not so”* (P19).

*“The definition inside the hospital… hum… it’s still misunderstood, that is, the definition inside the hospital is of the patient in the throes of death, here in oncology it’s not so, in oncology it’s indeed well understood that palliative care is all we do without intention to cure, as this concept isn’t well understood in the hospital, then, when you put palliative care in the medical record it can mean that the patient is dying and that we’re not going to do anything else”* (P26).


#### 3.2.2. Health Policy

The physicians demand technical assistance from experts who know PC in the control bodies. There are professionals who agree that it is important there is at least the written document and that PC is part of the Public Health Ministry rule. Political commitment is not always found, mainly as regards to implementing and financing what is set forth in the Ministerial Agreement.
*“…if we don’t have technicians in the Ministry who understand what palliative care is, hardly can we implement or launch a different project or a vision on how to treat the patients”* (P1).


One of the steps forward made is the elaboration and publication of the Clinical Practice Guide, the same that is known and used by many of the interviewed professionals. However, they agree on the weakness of the health system since many of the drugs they recommend are not sold in the country or are difficult to get, like morphine, which is not yet available in Ecuador in its oral presentation, but only for its parenteral administration route, although the supplies are not regular. In addition, there are opioid restrictions for access to the patients cared for at their homes.
*“…it’s also a problem, then also a little on the politics, right?, that they don’t go along exactly and it’s true that it’s incomprehensible for me, why is it that oxycodone is here and morphine is not?, not in its oral presentation…” “Because when I arrived at the hospital we didn’t have morphine, now we do, we already have the four or five basic opioid medications, it’s not all or far from it”* (P10).


Another of the identified deficiencies is the time devoted in consultations to patients, time which, on occasions, is insufficient to fill in the palliative care clinical history.
*“… I’d love to be able to have all my palliative care patients with their clinical histories as set out by the Health Ministry with their scales and evaluations, everything; but, physically we don’t have enough time”* (P26).


#### 3.2.3. Professional Activities

One of the weaknesses identified is the lack of professionals both trained and qualified in PC, as well as the scarce research materials on the subject matter. The participants consider that research on PC is important to disseminate and consolidate PC in Ecuador. Likewise, continuous training is necessary, although it is observed that there is yet no express interest on the part of the medical community. There are no associations of professionals linked to PC, which could contribute to developing the discipline, nor any support from institutions for such development.
*“…palliative care is a priority from the human point of view. Medicine has become very positivist, I mean, it only looks at the technical part and the human part which is part of each profession has been left aside and I wish that’s a wicked way for people to start reflecting and changing”* (P1).

*“I made events for the generalist, for the specialist, I even gave an international congress a whole day for palliative care, people came from Argentina, from Spain, they came to help us here and gave a course on palliative care, how many attended?, not even thirty, then?”* (P4).


#### 3.2.4. Health Services

There are no PC units or adequately trained professionals. Currently, there is not a single specific unit where patients who require PC are cared for, although they may exist, but they do not follow the required structure. Through the information collected in the interviews, we verified the lack of unified criteria to refer the patients, which constitutes a risk mainly when managing concepts like therapeutic proportionality and therapeutic obstinacy. This lack of definition has determined that, in one of the main third-level public hospitals, it is only the oncologists who refer the patients to PC, not other specialists.
*“The consultation has been dehumanized, because they’re more interested in the number of patients they see than in the quality of care, then, they’re patients they have to see in between ten and fifteen minutes and that only implies this, then, they never come back to see, they touch him, they examine him, if that’s the common practice in the consultation nobody’s worked on that palliative care is”* (P1).

*“Many people are going to say that they have services and units or that they provide palliative care and when we meet and we’re almost always the same we meet to talk about this in the country, we realize it’s not like that. …They may exist, even, there’s a sign in one institution, but there’s not a single person that is first, trained in palliative care and devoted in exclusive hours to caring for this type of patients and their families”* (P25).
*“…all of us oncologists who somehow have approached palliative care both as palliative chemotherapy and as situations of exclusive symptomatic palliative care, we somehow provide palliative care with training in courses, congresses, that kind of things. But indeed, it is super deficient, there’s not enough structure, there’s not enough home visits. Because the optimum thing would be that we could manage this in an outpatient and not hospital modality unless there were refractory symptoms or which really need hospitalization, but there are not enough doctors for outpatient treatments nor enough training to do so in the hospitals either, nor the infrastructure, this is a hospital for acute not chronic patients, therefore, the patients in palliative care state have a complex hospital stay because somehow what we want is being able to refer him to a chronic palliate care unit”* (P28).



Following with the interdisciplinary work, the participants state that there is no teamwork, mainly during hospitalization, where the specialists act independently. In addition, the fear that the word “palliative” awakens among the patients and their relatives is an obstacle for the specialists to send their patients to PC units. For this reason, some physicians avoid mentioning it and others prefer to call it differently.
*“They’ve changed the name of palliative care, because people felt very uncomfortable when going to palliative care. Because they said, if they mentioned palliative care as it is they said: no, it’s that I’m dying you can give me more chemotherapy, you can give radiotherapy you can’t send me to palliative care. Then, its name was virtually changed… I don’t remember right now… I believe they put it as a collaboration service or something like that, but they gave it a slightly different name so that people could go with fewer prejudices to palliative care”* (P20).


PC has started to develop in the biggest urban centers, but it has not yet reached rural areas or smaller cities in Ecuador, reason for which the participants recognize the need to strengthen these services and to disseminate them across the country.
*“In Ecuador we have a lot to do, a lot to do because we only have palliative offers in the big cities like Quito, Guayaquil, Cuenca, Ambato, and in another town I saw small cells, in Loja they have something, it’s only province capitals and not all… they don’t have in the centers… the sub-centers which would be the centers where they could help people who have no access, the staff is not prepared. So the rural doctor should know palliative care”* (P22).


On the other hand, weaknesses are also identified in the home visit, which should be strengthened as regards the team and medications supply.
*“The health center doesn’t solve what it has to solve and those who have social security home visit don’t solve it for them either, no, it’s that I don’t have this medication…”* (P16).


#### 3.2.5. Development of Palliative Care

In Ecuador, PC began in the 1990s, but it is in the last five years that it exhibits a peak in its development, partially due to the development of the Ministerial Agreement of 2011 in which PC was incorporated into the National Health System of Ecuador.
*“Let’s see, in Ecuador palliative care is a discipline which is only now showing its nose, not because it’s never existed, it’s existed since more or less twenty years if not a little more”* (P16).

*“I believe that in 2011 this had an inflection point, when finally the Public Health Ministry published a ministerial agreement… until that year what we had realized was that in the Public Health Ministry they simply didn’t talk about palliative care, there was no awareness of that”* (P30).


Furthermore, from that moment on is that the knowledge related to PC are included in the Family and Community Medicine post-graduate studies and that the paradigm solely centered on curing the disease is changed by including the PC perspective. On the other hand, it was observed that the new classes of trained physicians have veered and got closer to the PC approach.
*“It’s something totally new in the system… Totally new in the system which breaks any paradigm because this health system has always been focused on curing diseases…”* (P31).

*“Here there’s been a flexibilization to a certain extent in the institution which has allowed us to have a specific area of palliative care with pain management, with an external consultation, it has two offices, one procedures room, one meeting room”* (P25).


#### 3.2.6. Recommendations

From the perspective of the interviewees, the vision of PC improvement in Ecuador lies in the training of multi-disciplinary teams which work collaboratively and interdisciplinarity in order to try to transcend the disease beyond the soma, the psyche, the social and family environment and, particularly, in primary health care.
*“I’d like to see engaged people, that is, people who likes to do, that is, committed to help, then, hum… I’d like everything in the ideal team, I’d like three coworkers, I’d like the nurses, not one, I’d like three nurses, I’d like let’s say two psychologists for follow-ups, a first-time consultation, I’d like the auxiliary nurse to make… how do you say it… to make the cleansing there, skin care. People who are training right there, who see the reality of what you do, being able to give educational talks”* (P27).


To this end, they propose training oriented to PC in order to raise awareness of the philosophy which emanates from this specialization and to know how to manage pain. Among the skills to be trained, empathy and decision-making are also identified, as well as managing the most common and most complex signs and symptoms.
*“For them to refer the patients to us, for them to trust us”* (P11).

*“Coming to what’s most important: the patient, with adequate communication”* (P2).


Among the proposals derived from this research is creating the specialization, which would contribute to the full development of PC, as well as periodic and ruled trainings.
*“The next most important step is the master degree”* (P9).

*“Creating scholarships for the doctors who already have titles or physicians who are interested in palliative care training, some agreement could be made with some university and training more people in palliative care, then, having trained people I believe they wouldn’t fear facing this situation, because it’s the fear of not knowing what to do, what to say, that’s why, they’d better not face it”* (P8).


On the other hand, the need was stated for the creation of intra- and extra- hospital PC services, with the appropriate infrastructure to satisfy the care demand. Providing the necessary supplies, apart from an atmosphere of harmony and light, which favors peace in the patients and their relatives.
*“Creating the physical space for the therapy, palliative team units and two, creating the palliative care service, then, the time I have my physical space one or two beds and it’s me that’s going to manage my two-bed service, I’m going to start growing and structuring, then, I believe that’s what’s needed and it’s begun to be done here, but, realizing that it’s very important to start with the physical space and the team that’s going to manage that”* (P6).


## 4. Discussion

The aim of this study was to analyze the situation of PC in Ecuador from the perspective of the physicians. The current demographic and epidemiological changes evidence the insufficient number of PC professionals faced by Ecuador and by the whole world [19,20]. The increase in the number of non-transmissible chronic diseases, added to the aging of the population, calls for a strengthening of the care provided to the patients, starting mainly at the first level [21,22]. To attain this objective, it is indispensable to structure academic programs capable of giving answers to the needs of the system [23].

The limited training in PC of the health professionals is not an isolated fact in Latin America since only two countries, Cuba and Uruguay, include PC in their pre-graduate courses as a mandatory discipline in all their Medical Schools. Six Latin American countries do not integrate PC in their pre-graduate training programs and the other eleven offer PC either as a mandatory or optional discipline in some of their Medical Schools. Ten countries offer PC post-graduate programs for physicians (Argentina, Brazil, Colombia, Cuba, Costa Rica, Mexico, Panama, Paraguay, Uruguay and Venezuela) [24]. There is no official training on PC in Ecuador, but it is a known fact that five post-graduate programs have included it and that several ones have incorporated care for the end-of-life in their curricula as can be seen from the results found in this research study, results which are in line with those from other papers by other authors who already signaled the need for training the professionals from the University pre-graduate courses [24,25,26].

It is known that, when sustained in time, health and education policies directly contribute to improving the quality of health care and, consequently, the health of the population [25]. Hence, the participation of the public politics is fundamental to make PC effective as a human right [27,28,29]. In consonance with these claims, de Lima points out that, for PC strategies to be effective, they must be part of the National Health System and be covered by programs which allow providing care at all levels. However, the Pan American Health Organization (PAHO) does not establish any indicator that monitors the development of PC [30], nor has it executed any kind of surveillance so far. Despite the barriers identified in this study for this implementation, the implication of Politics is necessary for the development of PC in Ecuador, since the reality is that, according to studies from the PAHO, the incidence of cancer will increase by 67% in the American continent by 2030, adding to this the fact that it is often diagnosed in advanced stages; then, a clear PC strategy is needed [31].

The update conducted by Berterame et al. [32] on the use of morphine in the world evidences a substantial increase in consumption, although it remains low in Africa, Asia, Eastern Europe and Latin America. The reasons adduced are related to the lack of education on the use of opioids, to the existence of cultural barriers like opiophobia, and to regulatory and economic impediments, facts already verified with the results of this study. In accordance with that, Ortiz-Prado et al. state that the policies must be developed in a complementary manner with the economic strategies to realize them [33].

Another fact evidenced in this study—and which has already been described by several authors—is the lack of PC training in the generalists, who are in charge of the first care level or, in the case of Ecuador, who finish the mandatory year of Rural Medicine [34,35,36,37]. The aforementioned must be considered as an alert, if also considering what Ury, Reznich and Weber [38] point out about the fact that 80% of the deaths occur in health institutions, under the responsibility of generalists, internists or family doctors, who have gaps in their training, mainly as regards pain management, communicating bad news and decision-making. Therefore, it is essential to prepare the physicians in the treatment of advanced, chronic and incurable diseases. They must be qualified in the management of communication, in the adequate treatment of pain and in the use of opioids, as recognized by the physicians participating in this study, in consonance with several authors [39]. It is indispensable that the generalists be able to determine the PC needs of the patients and of their families, as well as to promptly identify the moment to refer the patient to the PC specialized team.

It is widely accepted that PC knowledge is not enough without a planned effort to expand the practice and to improve the health services [27,40,41,42]. As can be verified, since 2012 PC services have increased, although a certain prevalence is still observed in the private sector in Ecuador. The individuals with palliative care needs generally are identified at the third care level, where the palliative care model must be initiated under the responsibility of a highly specialized team. However, the first contact many patients have with a physician is at the first care level, reason for which it should be strengthened. Similar to Weissman and Meier [43], we recommend the implementation of systems which help the generalists to identify the patients who need PC to design care processes for the patients and their families, so that, if they see patients with some level of complexity, they receive support from the second care level or, in very complex cases, they can refer them to the third-level hospital.

This study is not exempt from limitations. Among them is the fact that it was limited to the urban areas of four Ecuadorian cities. However, the differential nuances that could occur in those places were approached by the participants, such as the poverty levels of the rural areas and in the cities with less population, as well as the difficulty in accessing health care or to move to urban areas to receive assistance or to buy medications that are only available in the hospitals.

## 5. Conclusions

It is necessary to maintain the implementation path of PC in Ecuador. It is essential that PC is included in the medical cross-training of the country, as well as to increase the number of services where PC are provided with optimum quality standards guaranteed by a multi-disciplinary team trained in the subject matter. Likewise, it is necessary to drive research that sustains the practice, as well as to provide material and human resources which allow for the provision of services, the effective support from Politics and from health institutions being required.

## Figures and Tables

**Table 1 ijerph-17-03906-t001:** Scripts of the interviews.

- What do you think of using opioids for pain management? - What is the physicians’ attitude towards end-of-life? - How is the relation with the patient like, for example, when delivering bad news? - In general, how must the physician act in the treatment of a patient subjected to PC?
- What is your perception about inter-disciplinary and multi-disciplinary work in PC? - What needs do you think exist as regards the development of PC in Ecuador? - Do you think these needs are covered? If they are not, which are the obstacles? - What is your perception about opioids use, availability and accessibility in Ecuador?
- Imagine that, on any given day while in your professional practice, you could choose an “ideal situation” for the development of PC in Ecuador (as if it was a “dream”). Which would that “dream” be for you? - Do you think that that “dream” you set out can be attained in the near future? How can it be attained? - What recommendations do you have for the development of PC in Ecuador?
- Do you have or have had any experience with patients that you would like to share? - Would you like to add anything else?

**Table 2 ijerph-17-03906-t002:** Sociodemographic data of the interviewees.

Atlas Ti Code	Interview Order	Age	Gender *	City **	Specialization	Years of Professional Experience	PC Training	Membership to PC Associations
P1	1st	57	F	Q	Intensive care	22	PhD	No
P2	2nd	57	M	Q	Anesthesiology	25	Courses	No
P3	3rd	52	M	Q	Anesthesiology	15	Courses	No
P4	4th	50	M	Q	Oncology	21	Courses	No
P5	5th	34	F	Q	Internal medicine	5	Courses	Yes
P6	6th	46	M	Q	Oncology	14	Courses and during specialization	No
P7	7th	57	F	Q	Family medicine and Public Health	30	Courses and Conferences	No
P8	8th	35	M	Q	Family medicine	2	During specialization	No
P9	9th	48	F	Q	Family medicine	17	Courses PhD Master’s degree	Yes
P10	10th	41	F	Q	Oncology Internal medicine	8	Courses and During specialization	No
P11	11th	44	F	L	Anesthesiology	8	Courses Remote master’s degree (in progress)	Yes
P12	12th	43	M	L	General and oncological surgery	12	–	No
P13	13th	47	F	L	Pediatric onco-hematology	14	Conferences	No
P14	14th	40	F	L	Onco-hematology	6	Conferences	No
P15	15th	36	M	L	Oncology	11	Courses and During specialization	No
P16	16th	48	F	Q	Internal medicine	14	Courses Conferences Master’s degree	Yes
P19	17th	37	M	C	Master’s degree in health Management	10	Courses Conferences Workshops	No
P20	18th	56	M	C	General medicine	30	Courses Conferences	No
P21	19th	55	M	C	Family medicine	30	During specialization	No
P22	20th	63	F	G	Anesthesiology	38	Master’s degree	Yes
P23	21st	36	F	G	Oncology	6	Master’s degree	No
P24	22nd	49	F	G	General medicine	19	Courses Conferences	No
P26	24th	60	M	L	Oncology	30	Conferences	No
P27	25th	46	M	L	Oncology	7	Courses Conferences	No
P28	26th	40	F	Q	Oncology	7	Courses	No
P30	27th	45	F	Q	Oncology and Internal medicine	10	PhD	No
P31	29th	– ^a^	F	Q	master’s degree in public health	– ^a^	– ^a^	– ^a^
P25	30th	52	M	Q	Anesthesiology and pain management	20	Courses	No

^a^ = The participant did not fill in the information; * gender: M (male), F (female); ** city: C (Cuenca), G (Guayaquil), L (Loja), Q (Quito).

**Table 3 ijerph-17-03906-t003:** Relation of the identified themes.

Theme	Description
(a) Training	PC training ^1^ at graduate and post-graduate levels, as well as on the opportunities of official accreditation by the Secretariat of Higher Education in Science, Technology and Innovation (*Secretaría de Educación Superior, Ciencia, Tecnología e Innovación*, SENESCYT) of those post-graduate degrees obtained abroad. The existence of PC training programs was also analyzed.
(b) Health policy	The knowledge of the participants about the health policy which regulates PC in Ecuador, including legal aspects involving care, distribution policies and access to medications (mainly opioids), academic policies and governmental resources allocated to this specialization.
(c) Professional activities	The knowledge of the professionals about the different national PC associations and care networks was explored and determined. The research activities developed in the country were investigated, as well as the possible collaborations maintained with international institutions and bodies.
(d) Health services	It refers to the socio-sanitary care levels in Ecuador. The care level includes community health centers, home care and hospices. The second care level includes the services and/or exclusive units in the second-level hospitals. The third care level includes the services and/or exclusive units in third-level hospitals. For the second and third levels, the hospital support teams were also evaluated. Finally, the existence of multi-level teams was determined for all the care levels.
(e) Development of PC	Details were investigated on the perception of the development of the PC services and care in Ecuador.

^1^ PC (palliative care).

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
