# Peer review of "Knowledge, Attitudes and Expectations of Physicians with Respect to Palliative Care in Ecuador: A Qualitative Study"

_ijerph, 2020, doi:10.3390/ijerph17113906_

Round 1

Reviewer 1 Report

Comment 1: the authors drew samples from different medical subspecialities. There is greater scope to compare knowledge, attitude and practice of PC among different specialist physicians

C2: Table 2 suggests level of training of physicians in PC but does not indicate knowledge inadequacies. What is the authors' inference?

C3: The need for incorporating PC in medical curricula is appropriately highlighted and so is the unavailability of opioids. 

C4: What is hospice-type??? Line 70

c5: Need to specifiy whether Ecaudor is a low or mid-income country? Line 19

C6: English expressions could have been improved a lot.

Author Response

REVIEWER 1

Dear reviewer,

We would like to thank you your revision that helps us to improve our manuscript and make it suitable for publication. We appreciate your time.

Comment 1: the authors drew samples from different medical subspecialities. There is greater scope to compare knowledge, attitude and practice of PC among different specialist physicians

Response: Thank you for this comment. In fact, our intention was to draw a wide picture from different point of views.

C2: Table 2 suggests level of training of physicians in PC but does not indicate knowledge inadequacies. What is the authors' inference?

Response: Thank you for this observation that we find very interesting. We asked about training in palliative care to explore differences among professionals. Nevertheless, we are aware that not always training is synonym of skills and abilities so we compared this idea in the discussion section (Please, see line 386)

C3: The need for incorporating PC in medical curricula is appropriately highlighted and so is the unavailability of opioids. 

Response: Thank you for this observation that is highlighted since this is basic to provide a proper service.

C4: What is hospice-type??? Line 70

Response: Thank you for this question. We changed the statement to make it more understandable: Likewise, there are hospice-type [13] community institutions called hospice for when home care is not possible and hospital admission is not advisable.

c5: Need to specifiy whether Ecuador is a low or mid-income country? Line 19

Response: Thanks for this query. Now, you can find in the text that Ecuador is a mid-income country (Please, see line 19).

C6: English expressions could have been improved a lot.

Response: Thank you for this observation. The manuscript was reviewed and edited by a professional service.

Reviewer 2 Report

This is an interesting exploration of physicians' views of palliative care in the given context. The paper is strong, but can be improved in a couple of areas:

  1. The text needs a final proofreading to rectify minor errors.
  2. Page 1, line 37, it reads 'mourn process'; did you mean 'mourning'? (not a process in this case).
  3. The background section sets the scene, yet can be strengthen with more literature on palliative care - its development and qualities. The authors refer to the principles of universal health and palliative care as a duty; these claims need more substance. I wonder if the authors have read the Oxford Handbook of palliative care. This must be in its fifth edition at the minute and provides insights about the growth of palliative medicine and care.

  4. Page 3, line 97: can you give more details about the researcher's training?
  5. Regarding geography, are there any differences contextually and socially between the four cities. Could this also be playing a part in  contrasting physicians' views between the cities? 

Author Response

REVIEWER 2

Dear reviewer,

We would like to thank you your revision that helps us to improve our manuscript and make it suitable for publication. We appreciate your time.

This is an interesting exploration of physicians' views of palliative care in the given context. The paper is strong, but can be improved in a couple of areas:

  1. The text needs a final proofreading to rectify minor errors.

Response: Thank you for this observation. The manuscript was reviewed and edited by a professional service.

  1. Page 1, line 37, it reads 'mourn process'; did you mean 'mourning'? (not a process in this case).

Response: Thank you for this correction. This was changed following your recommendation.

  1. The background section sets the scene, yet can be strengthen with more literature on palliative care - its development and qualities. The authors refer to the principles of universal health and palliative care as a duty; these claims need more substance. I wonder if the authors have read the Oxford Handbook of palliative care. This must be in its fifth edition at the minute and provides insights about the growth of palliative medicine and care.

Response: Thank you for this question. This study was the first author´s research thesis and she read much literature about palliative care during that period. We included some ideas of that book in the manuscript but supported by authors that included ideas in their publications.  An example is Pastrana, 2012. However, we decided to include some ideas in the manuscript and add the reference suggested. (Please, see lines 63 to 68 and reference [10], lines 438-439).

  1. Page 3, line 97: can you give more details about the researcher's training?

Response: Thank you for this query. The researchers had the opportunity to receive training in quality methods in order to develop the project. Interviewer was trained about phases of a qualitative interview: welcome/ introduction, consent process, demographic questions, icebreaker, main interview questions, probes, a review process, and conclusion. Besides, she received instructions about how to handle the interview to achieve the purpose of the research. This information was included in the manuscript.

  1. Regarding geography, are there any differences contextually and socially between the four cities. Could this also be playing a part in  contrasting physicians' views between the cities? 

Response: Thank you for this timely question. Cities were chosen because that cities are the highest ones in the country. Nevertheless, politics have to be followed  National Country. We did not find difference among professionals but we did find related to availability of resources. Quito and Guayaquil had more resources, and Loja and Cuenca less. This idea was included in the manuscript.

Please see lines 85-89: Quito and Guayaquil are the most important cities in the country, and Loja and Cuenca are smaller cities. This provided us a wide vision of the situation of PC care in distinct areas around the country. PC care in these cities is similar because physicians follow national politics but access to resources is quite different, having less availability of them in Cuenca y Loja. Besides, these four cities show highest mortality rates in the country.
